# Navigating Brain Organoid Maturation: From Benchmarking Frameworks to Multimodal Bioengineering Strategies

**DOI:** 10.3390/biom15081118

**Published:** 2025-08-04

**Authors:** Jingxiu Huang, Yingli Zhu, Jiong Tang, Yang Liu, Ming Lu, Rongxin Zhang, Alfred Xuyang Sun

**Affiliations:** 1State Key Laboratory of Oncology in South China, Sun Yat-sen University Cancer Center, Guangzhou 510060, China; huangjx@sysucc.org.cn; 2Signature Research Program in Neuroscience and Behavioral Disorders, Duke-NUS Graduate Medical School Singapore, 8 College Road, Singapore 169857, Singapore; yl.zhu@nus.edu.sg; 3Singapore Bioimaging Consortium, Agency for Science Technology and Research (A*STAR), Singapore 138667, Singapore; tang_jiong@imcb.a-star.edu.sg; 4Department of Pharmacology, Nanjing University of Chinese Medicine, Nanjing 210023, China; liuyang@njucm.edu.cn; 5Jiangsu Key Laboratory of Neurodegeneration, Department of Pharmacology, Nanjing Medical University, Nanjing 211116, China; lum@njmu.edu.cn

**Keywords:** brain organoids, maturation, bioengineering strategies, microenvironment modulation, organoid vascularization, AI driven platform

## Abstract

Brain organoid technology has revolutionized in vitro modeling of human neurodevelopment and disease, providing unprecedented insights into cortical patterning, neural circuit assembly, and pathogenic mechanisms of neurological disorders. Critically, human brain organoids uniquely recapitulate human-specific developmental processes—such as the expansion of outer radial glia and neuromelanin—that are absent in rodent models, making them indispensable for studying human brain evolution and dysfunction. However, a major bottleneck persists: Extended culture periods (≥6 months) are empirically required to achieve late-stage maturation markers like synaptic refinement, functional network plasticity, and gliogenesis. Yet prolonged conventional 3D culture exacerbates metabolic stress, hypoxia-induced necrosis, and microenvironmental instability, leading to asynchronous tissue maturation—electrophysiologically active superficial layers juxtaposed with degenerating cores. This immaturity/heterogeneity severely limits their utility in modeling adult-onset disorders (e.g., Alzheimer’s disease) and high-fidelity drug screening, as organoids fail to recapitulate postnatal transcriptional signatures or neurovascular interactions without bioengineering interventions. We summarize emerging strategies to decouple maturation milestones from rigid temporal frameworks, emphasizing the synergistic integration of chronological optimization (e.g., vascularized co-cultures) and active bioengineering accelerators (e.g., electrical stimulation and microfluidics). By bridging biological timelines with scalable engineering, this review charts a roadmap to generate translationally relevant, functionally mature brain organoids.

## 1. Introduction

Human brain organoids—self-organizing three-dimensional structures derived from pluripotent stem cells—recapitulate defining features of the developing human brain that are inaccessible to conventional models [1]. Region-specific organoids have been generated mimicking the cortex [2], midbrain [3], and cerebellum [4], while integrating diverse cellular components, including neurons, astrocytes, oligodendrocytes, and other supporting cells, in spatiotemporally appropriate sequences [5,6,7]. In some cases, these organoids develop fluid-filled cavities resembling ventricles, which are crucial for nutrient transport and waste removal, further enhancing their physiological relevance [8,9]. Notably, brain organoids are capable of exhibiting electrical activity and forming synaptic connections, releasing intrinsic neurotransmitters, thereby demonstrating functional characteristics akin to those of the human brain [10,11,12,13]. Collectively, these advantages present a valuable avenue for probing the intricacies of human brain development and the pathophysiological processes of disease in vitro, a feat that traditional two-dimensional cultures and animal models struggle to replicate [2,13,14,15]. By bridging these gaps, brain organoids establish an unprecedented human-relevant platform for probing neurodevelopment, modeling pathogenesis of neurological disorders, and performing clinically predictive drug/toxicity screening [16,17,18].

Despite significant advancements, brain organoid maturation remains fundamentally constrained at fetal-to-early postnatal stages—even after extended culture (>100 days). This developmental arrest precludes modeling of adult neurological disorders (e.g., Alzheimer’s disease requiring mature amyloid-β processing) and compromises drug screening validity due to immature pharmacodynamic responses [19,20,21,22]. Critically, the maturation and functionality of key supportive cell types, particularly astrocytes, remains a major bottleneck. This includes their failure to robustly form essential structures like the glia limitans and a fully functional blood–brain barrier (BBB) in current organoid paradigms [23,24,25,26]. Furthermore, modeling the diverse reactive states of astrocytes (e.g., neuroprotective vs. pro-inflammatory) and their specific functional roles in critical processes, such as amyloid-β clearance, edema regulation, and neuroimmune modulation, remains largely unvalidated and technically challenging [27,28]. Compounding these limitations, the field lacks standardized maturity metrics, with current assessments varying from fragmented molecular markers to isolated electrophysiological readouts. This methodological heterogeneity impedes cross-study comparability and protocol optimization. Furthermore, prolonged culture often induces hypoxia-driven central necrosis, diminishing cellular diversity and structural integrity [29,30]. These microenvironmental stresses further impede the functional maturation of non-neuronal cells, especially astrocytes, hindering their ability to establish complex barrier functions and exhibit physiologically relevant reactivity. To resolve the oxygenation and nutrient diffusion issue during the long-term culture of organoids, and to accelerate the developmental progression toward more mature states, researchers are actively pursuing different strategies, such as more sophisticated culture systems, bioreactors, and bioengineering approaches that can better mimic the environment in vivo, and to produce higher quality organoids.

This review summarizes these advances through two interconnected lenses: (1) standardization of maturity benchmarks, proposing multimodal evaluation frameworks to unify field-wide assessments, and (2) a comprehensive summary of innovative methodologies conducive to enhancing the long-term culture environment and fostering the maturation of brain organoids, from microenvironmental tuning and mechanotransduction modulation to supportive cellular integration. By critically evaluating their synergistic potential, we chart a translational roadmap for generating brain organoids that recapitulate adult-stage functionality—bridging the critical gap between in vitro models and human neuropathology.

## 2. Multidimensional Framework for Assessing Brain Organoid Maturity

The assessment of brain organoid maturity is a critical aspect of understanding their development and functionality, as these organoids serve as models for human brain development and disease. The selection of evaluative dimensions encompasses a spectrum of structural, functional, and biological characteristics, each reflecting the profound complexity inherent to the human brain. Among the key dimensions are morphological attributes, such as the meticulous organization of neuronal layers and the discernible presence of distinct cell types, including the emergence and spatial distribution of glial populations, which are indispensable for emulating the architectural finesse of the brain. Equally paramount are functional assessments, including electrophysiological properties and glia-driven homeostatic processes (e.g., glutamate uptake and phagocytose synaptosomes) alongside BBB functionality, which yield critical insights into the organoids’ capacity to recapitulate integrated neural tissue physiology. Furthermore, the assessment must encompass biochemical/molecular markers associated with pan-cellular maturation, notably the expression of specific genes and proteins that signify neuronal differentiation, non-neuronal differentiation, synaptic formation, and BBB formation. Accurately evaluating the maturity of brain organoids using scientifically validated methods is a critical and indispensable step in individual studies, as it directly impacts the repeatability and reliability of experimental outcomes and their translational relevance. Within the broader field of brain organoid research, continuous refinement and innovation in assessment strategies—through the integration of multidisciplinary approaches—are essential to optimize experimental efficiency, reduce costs, and enhance reproducibility. These advancements not only improve model fidelity but also drive the development of standardized protocols, which are pivotal for advancing the field toward clinical and pharmaceutical applications. We will outline the well-established methodologies for assessing brain organoid maturity, including structural architecture, cellular diversity, functional maturation, and molecular signatures (see Figure 1).

Structure architecture. Structural maturation of brain organoids is defined by the progressive acquisition of anatomically layered cytoarchitecture, functional synaptic connectivity, and region-specific molecular identities—features essential for recapitulating human cortical development. Critical benchmarks may include cortical lamination validation: SATB2 demarcates upper-layer (II-IV) populations, while TBR1 identifies deep-layer (VI-V) neurons, and CTIP2 expresses in deep layers, especially layer V [11,19,31]. Beyond laminar organization, the development of essential barrier structures should be evaluated in some cases: formation of the glia limitans externa (visualized via aquaporin 4 expressing astrocyte endfeet alignment at organoid periphery) in glia-enriched cortical organoid transplanted in mouse brain [32] and rudimentary blood–brain barrier units (identified by CD31^+^ endothelial tubes ensheathed by PDGFRβ^+^ pericytes and contacting GFAP^+^ astrocytic processes) [26,33]. Complementarily, synaptic maturation is quantified through presynaptic synaptobrevin-2 (SYB2) localization in synaptic vesicles, while postsynaptic integrity is confirmed by PSD-95 clustering in dendritic spines [34]. Regional identity is established via combinatorial transcription factor signatures: FOXG1 defines forebrain identity, with PAX6 specifying dorsal telencephalic domains and NKX2.1 characterizing ventral/ganglionic eminence derivatives [10,11]. These structural and regional benchmarks are primarily visualized through immunofluorescence (IF) and immunohistochemistry (IHC), often enhanced by confocal microscopy for three-dimensional cytoarchitectural analysis of both neuronal and non-neuronal components. Ultimately, ultrastructural validation is achieved via electron microscopy (EM), which resolves synaptic vesicles and postsynaptic densities, tight junction complexes between endothelial cells, and polarized astrocyte endfeet abutting vascular-like structures at nanoscale resolution to confirm functional maturation [20,32,35,36].

Diversity of cell types. Precise characterization of neural populations in brain organoids relies on cell-type-specific molecular markers, validated through complementary analytical approaches. General neuronal markers, including NEUN (RBFOX3) and βIII-tubulin (TUBB3), broadly identify neuronal lineages, while maturity-stage markers distinguish developmental states: DCX and NeuroD1 label immature neurons, whereas MAP2 demarcates mature neuronal populations [37,38,39]. Critically, neurotransmitter identity further subcategorizes neurons—glutamatergic excitatory neurons express VGLUT1, while GABAergic inhibitory neurons are identified through GABA synthesis enzymes (GAD65/67), vesicular transporters (VGAT), or GABA itself [38,40]. Beyond neuronal characterization, astrocytes are recognized by GFAP and S100β expression and oligodendrocytes by myelin basic protein (MBP) and O4 surface antigen [5,15]. To spatially resolve these populations, IF and IHC remain essential; conversely, for quantitative profiling of cellular heterogeneity, fluorescence-activated cell sorting (FACS) provides high-throughput single-cell resolution. This multimodal marker framework enables comprehensive mapping of organoid development across structural and functional dimensions.

Function maturation. The examination of functionally active neurons, connected neural networks, glial homeostatic functions, and neurovascular barrier integrity in brain organoids, is essential to monitor and evaluate the maturity and physiological relevance. Classical electrophysiological techniques, such as patch clamp, afford high temporal resolution of neural activity with organoids, albeit with limited spatial resolution for whole-organoid assessment [5]. Calcium imaging utilizes fluorescent indicators to visualize dynamic calcium transients within neurons and increasingly in astrocytes using GLAST-promoter driven GCaMP reporters, which serve as a reliable proxy for neural and glial activity [41,42,43]. This technique excels at mapping the spatial patterns of activity across populations of cells or brain regions. However, calcium imaging offers relatively minimal temporal resolution due to the inherently slow kinetics of calcium signaling itself (the transient rise and decay) and practicality from the finite acquisition speed of optical imaging systems [14,41,42]. Multielectrode arrays (MEAs) emerge as the novel standard for electrophysiological assessment, recording synchronized neuronal network activity, including γ-band oscillations and spontaneous action potentials [44,45].

Molecular and Metabolic Profiling. Single-cell RNA sequencing (scRNA-seq) serves as a cornerstone for evaluating structural maturation in brain organoids by resolving cellular heterogeneity through transcriptome-wide profiling of individual cells, enabling precise quantification of neuronal/glial proportions and developmental states [46,47]. Building upon this foundation, pseudotime analysis computationally reconstructs temporal trajectories to model neurodevelopmental dynamics, thereby quantifying maturity progression across inferred timelines [48,49]. Complementarily, spatial transcriptomics maps gene expression within intact tissue architectures—bridging single-cell resolution with topographical context—to reveal region-specific maturation gradients (e.g., cortical layer formation) inaccessible to dissociative methods. Furthermore, metabolomics provides critical functional insights by quantifying ATP production, lactate accumulation, and lipid droplet dynamics, directly assessing metabolic adaptation to microenvironmental stressors like hypoxia. Together, this multimodal framework—spanning transcriptional, spatial, and metabolic dimensions—delivers a systems-level understanding of organoid maturation [50].

## 3. Temporal Dynamics and Engineering Paradigms for Brain Organoid Maturation

### 3.1. Chronological Maturation in Brain Organoids Rationale

Organoids can be cultured for several months to several years. Setting aside considerations of temporal and labor costs, it stands to reason that the prolonged cultivation of brain organoids in vitro facilitates a deeper degree of ‘maturation’ within these constructs. This temporal dependency inherent to the maturation of brain organoids emerges as a pivotal characteristic, underpinning their utility in the modeling of human brain development and associated pathologies. A quintessential illustration of the intricate relationship between the complexity of cellular compositions and the formation of layers, juxtaposed with the duration of culture, is exemplified in dorsal forebrain organoids nurtured over a period of 20 months, capturing the essence of corticogenesis [34]. Here, radial glial progenitors materialize, succeeded by the progressive emergence of both deep and superficial glutamatergic neurons, alongside the generation and maturation of astrocytes [51]. Single-cell transcriptomic analyses have further substantiated that extended culture recapitulates transcriptional trajectories reminiscent of fetal brain development, marked by the sequential activation of neurogenesis, gliogenesis, and synaptogenesis [52]. Moreover, longitudinal observations of human brain organoids have unveiled that the maturation of neural and oligodendroglial lineages unfolds over months, with synchronized neuronal network activity and myelination patterns manifesting only after considerable durations of culture [53]. Yet, despite these advancements, challenges such as hypoxia-induced necrosis in non-vascularized organoids, cost, and contamination impose limitations on the practical extension of culture periods. Consequently, the majority of brain organoids in published studies have been cultivated within a timeframe of 120 to 150 days [2,15,35,54]. Progress in dynamic bioreactor systems and vascular co-culture models has alleviated these concerns, fostering stable maturation with enhanced metabolic support [55,56,57]. These findings underscore that temporal progression is indispensable for achieving organoid models capable of mimicking late developmental events, such as myelination or neurodegenerative pathology, which have not even been initiated in short-term cultures.

While extended culture durations permit gradual organoid maturation, this passive temporal approach remains resource-intensive and impractical for time-sensitive applications like drug screening. To accelerate brain organoid maturation, small-molecule compounds (e.g., growth factors and signaling pathway modulators) are widely used to steer neuronal differentiation or suppress off-target cell proliferation [21,52,58,59]. Despite their extensive application, these chemical interventions demonstrate limited efficacy in inducing mature characteristics. This fundamental shortcoming arises from their inability to mimic the dynamic biophysical and cellular interactions of native brain microenvironments.

### 3.2. Strategies to Enhance the Maturation of Brain Organoids In Vitro

To overcome the constraints of passive chronological maturation and chemical approaches, contemporary research focuses on multimodal bioengineering strategies that actively reconstruct neurodevelopmental niches. By recapitulating dynamic biophysical, metabolic, and cellular interactions of native brain microenvironments, these integrated approaches accelerate organoid maturation across multiple dimensions (see Figure 2). The following sections detail their implementation for generating electrophysiologically active, synaptically mature organoids within clinically relevant timeframes.

#### 3.2.1. Microenvironment Modulation

Vascularization of organoids. The establishment of functional vascular networks significantly enhances the supply of oxygen and nutrients while facilitating the elimination of waste, thereby mitigating hypoxia-induced necrosis and enabling the prolonged culture of brain organoids. For instance, Shi et al. elucidated that vascularized organoids, achieved through the co-culturing of human pluripotent stem cells with endothelial cells (ECs), exhibited remarkable survival for up to 200 days, accompanied by an increase in neuronal diversity and synaptic density [34]. In another approach, a distinct group directly induced endothelial differentiation by overexpressing the transcription factor human EST Variant 2 (ETV2) within pluripotent stem cells, culminating in the generation of vascularized brain organoids that thrived for an extended duration of 120 days, thereby enhancing the maturation of the organoids and even imparting features reminiscent of the BBB [26]. In in vitro culture, ECs enhance brain organoid maturation through multifaceted mechanisms. These extend beyond perfusion support and BBB mimicry [36,60] to include neurogenesis modulation by EC-secreted angiocrine factors [61,62] and metabolic–neuronal activity synchronization via EC-derived signaling molecules such as nitric oxide (NO) and brain-derived neurotrophic factor (BDNF) [56,57]. While the aforementioned strategies focus on in vitro vascularization, an alternative approach leverages transplantation into rodent hosts to achieve functional vascular integration. The transplantation of these brain organoids into rodent brains capitalizes on host vascularization, resulting in an impressive survival rate of approximately 233 days, alongside increased cell survival, maturation, and synaptic connectivity within the in vivo vascularized organoid [63]. Though primarily an in vivo methodology, this strategy provides valuable mechanistic insights applicable to refining in vitro vascularization techniques.Bioreactor. Rotational systems serve to enhance nutrient diffusion and replicate fluid shear stress, thereby promoting the self-organization of three-dimensional structures. Lancaster et al. were pioneers in this innovative approach, employing a spinning bioreactor to extend the complexity of organoids and prolong their culture duration to an impressive 300 days [2]. Other bioreactors, such as the orbital shaker, offer a low-shear environment, allowing organoids to be cultured with a ‘single house’ manner across various media or conditions [37]. However, a notable limitation of both the spinning bioreactor and the orbital shaker lies in their substantial consumption of media enriched with growth factors and essential compounds, as well as the requirement for extensive incubator space. In response to these challenges, the miniaturized spinning bioreactor, dubbed ‘SpinΩ’, was developed to reduce culture volume, thereby increasing throughput and reproducibility of the organoids while maintaining their viability for over 200 days [30].Microfluidic system. Microfluidic devices adeptly simulate the dynamics of blood flow, facilitating long-term culture with enhanced oxygen exchange and nutritional supply. Evidence has demonstrated that microfluidic devices, when integrated with brain extracellular matrix, significantly improve oxygen delivery to the core regions of organoids at day in vitro 120, thereby enhancing cell survival and reducing apoptosis, ultimately promoting both structural and functional maturation of the organoids [33]. Additionally, the innovative combination of brain organoids and microfluidic systems, referred to as ‘brain organoids-on-chips’, serves to refine culture conditions by minimizing shear stress, augmenting oxygen supply, and optimizing material exchanges, thus fostering the development of cortical layers, increasing organoid size, and enhancing electrophysical functions, all of which are essential for more accurately modeling brain development [64].Slicing organoid technique. In the realm of brain organoids cultivated within the confines of suspension culture, a notable expansion is observed, reaching diameters of 3 to 4 mm. Yet, within the depths of these burgeoning structures, the cells grapple with hypoxia, culminating in the formation of a necrotic core that imposes limitations on both growth and maturation. The innovative sliced organoid system, characterized by its disk-shaped configuration, unveils the interior to the external culture milieu, thereby circumventing the constraints of diffusion. This advancement not only mitigates cell death but also fosters the development of a more complete architecture and diverse cell subtypes over an extended culture period, approximately 150 days [65].Air–liquid interface culture. A growing body of evidence underscores its efficacy in alleviating hypoxia and enhancing cellular viability within the core of organoids [66,67]. A particular study explored a novel approach that marries the traditional slice culture technique with the air–liquid interface system, yielding remarkable improvements in survival rates and the maturation of morphology, marked by extensive axon outgrowths. Notably, this method enables the brain organoid culture to endure for as long as one year [67].Adhesion-Based Culture Platform. In the context of the adhesion-based culture platform, this methodology addresses hypoxia by diminishing organoid thickness and promoting astrocyte migration, as evidenced in the long-term adhesion brain organoid (LT-ABO) system. In essence, Xianwei Chen and colleagues executed a meticulous slicing of brain organoids once the characteristic layered structure had been established, typically after 70 to 100 days. These organoid slices were then cultured upon Matrigel-coated plates, facilitating adhesion and extending the culture duration to an impressive one year [31].

#### 3.2.2. Spatial and Structural Engineering Strategies

Mechanotransduction—the intricate process through which cells transmute mechanical stimuli into biochemical signals—stands as a cornerstone in orchestrating the structural and functional maturation of brain organoids. By emulating the in vivo mechanical cues, researchers are able to amplify the complexity, reproducibility, and physiological relevance of these organoids. In the ensuing discourse, we shall elucidate the spatial and structural engineering strategies through which the modulation of mechanotransduction expedites the maturation of brain organoids. It is pertinent to note that the ‘dynamic culture system’ previously referenced, which also partially facilitates organoid maturation via this mechanism, shall remain unexamined in this context.

Matrigel and the alternatives. The extracellular matrix (ECM) emerges as a pivotal architect in steering the maturation of brain organoids, furnishing biochemical, biomechanical, and structural cues that closely mirror the in vivo neural microenvironment [68,69]. The mechanical properties of the ECM directly influence neuronal differentiation and cortical layering. Matrigel, a basement membrane extract derived from mouse sarcoma, serves as an exogenous supplement of ECM. However, the implications of Matrigel on the architecture and cellular specification of brain organoids produced through various protocols remain a subject of contention [5,70,71]. The inherent complexity of Matrigel, coupled with its batch-to-batch variability stemming from compositional fluctuations, underscores its limitations in organoid culture. Recent strides in ECM engineering have concentrated on refining both natural and synthetic matrices as viable alternatives to Matrigel, aimed at augmenting neuronal differentiation, structural intricacy, and functional maturation.Decellularized brain tissue-derived ECM. Decellularized brain tissue-derived ECM presents a novel avenue, retaining region-specific biochemical signals—such as laminin, neurocan, and heparan sulfate—that are critical for neurogenesis, axon guidance, and synaptic formation. For instance, the application of human brain ECM (BEM) within microfluidic systems has been shown to significantly enhance cortical layering and electrophysiological activity in brain organoids when juxtaposed with Matrigel-based cultures at day 30 [33].Synthetic hydrogels. Synthetic hydrogels have also emerged as a promising frontier. Andrew et al. demonstrated that cortical organoids differentiated within neural cadherin peptide-functionalized gelatin methacryloyl hydrogels (GelMA-Cad) exhibited a greater frequency of spontaneous excitatory post-synaptic currents and action potentials in comparison to those cultivated in Matrigel, indicating that GelMA-Cad fosters the maturation of cortical organoids [72]. Other laboratories have formulated and employed hyaluronic acid-based matrices (HA-based matrices) to replicate the elastic modulus characteristic of brain tissue, thereby promoting the differentiation of neural progenitors into functional neurons. Notably, they discovered that HA-based matrices facilitate neuronal network formation by day 7 and 14, particularly in the presence of neighboring astrocytes [73].

In addition, 3D-Printed Microwell Platforms, eschewing the use of Matrigel or external signaling molecules, have been reported to successfully generate mature brain organoids exhibiting robust formations, such as wrinkling, folding, lumens, and neuronal layers. Particularly, organoids nurtured within microwell devices featuring high-resolution concave surfaces demonstrate a greater degree of maturation when compared to those cultured in T25 flasks at day 20 and day 45 [74].

#### 3.2.3. Stimulation-Based Strategies

The external electrical signal emerges as a pivotal force in the intricate tapestry of human tissue physiology and development. Numerous studies posit that the infusion of external stimuli during the nascent stages of brain organoid maturation may catalyze their evolution, fostering neural plasticity, neurogenesis, synaptogenesis, and the formation of various structures [43,75,76,77]. Notably, Li Xiaohong et al. elucidated that electrical stimulation (ES) significantly enhanced the differentiation and maturation of organoids, yielding superior cell viability and a robust functional electrophysiology. For instance, subcortical projection neurons (CTIP2+) manifested within the ES-treated organoids, a stark contrast to their scarcity in control specimens by day 40. Furthermore, an upregulation of genes associated with synaptic transmission, projection growth, and calcium channel activity was observed in the ES group by day 55 [44].

Beyond the realm of ES, magnetic stimulation, employing magnetoelectric scaffolds or magnetic nanoparticles, has been harnessed to modulate ion channel activity and orchestrate spatial organization within neural tissues, culminating in enhanced neuronal differentiation and network formation. Empirical evidence suggests that low-intensity magnetic fields can amplify calcium signaling and promote neurite outgrowth within organoid cultures [78].

Moreover, light stimulation, particularly through optogenetic activation or photo stimulation utilizing light-sensitive proteins, affords a precise temporal and spatial command over neuronal circuits. Recent investigations have illuminated that chronic patterned light exposure in photosensitive brain organoids can incite neural synchronization and elevate the expression of activity-dependent genes such as *c-Fos*, heralding an era of enhanced functional maturation [79].

#### 3.2.4. Integration of Supportive Cell Types

In the intricate landscape of the human brain, glial cells coexist alongside neurons, fulfilling pivotal roles in the orchestration of brain development, the maintenance of homeostasis, and the progression of disease [80]. Among these, astrocytes emerge as the most abundant cell type within the mammalian brain, engaging in seemingly passive yet essential functions within the central nervous system, such as the recycling of neurotransmitters and the regulation of synapse formation [23,81]. The factors secreted by astrocytes are of paramount importance, serving as critical signals for the formation and maturation of neural circuits [82,83]. Recent studies demonstrate that astrocyte-conditioned medium (ACM) enhances neuronal layer thickness and deep-layer cortical neuron density in forebrain organoids by days 60–90, indicating astrocyte-driven neural maturation [50]. While ACM provides soluble factors, it may not fully replicate the complex physical and bidirectional signaling present in integrated astrocyte–neuron–vascular co-cultures necessary for glia limitans formation or BBB induction.

Unlike neuroectoderm-derived neurons and astrocytes, microglia originate from mesodermal lineages. Consequently, standard brain organoids lack resident microglia [55,84,85]. Nevertheless, a growing body of evidence suggests that the co-culture of microglia with brain organoids significantly augments neuronal maturation within these constructs. In a notable study, Dong et al. successfully co-cultured brain organoids with primitive-like macrophages, thereby generating microglia-sufficient organoids, which demonstrated enhanced neurogenesis both phenotypically and functionally through the mechanism of cholesterol transfer [86]. Furthermore, other researchers have delineated an innovative approach that integrates microglia derived from human iPSCs into midbrain organoids, revealing that these microglia influence synaptic remodeling and elevate neuronal excitability within the midbrain constructs [87].

It is worth noting that while integrating supportive cells like microglia and endothelial cells holds promise, achieving functional maturation of astrocytes within brain organoids, particularly their ability to form critical structures and exhibit defined reactive states, represents a significant ongoing challenge that limits model fidelity. Current approaches often yield astrocytes that are present but lack the complexity and functionality observed in vivo.

## 4. Discussion

Over the past decade, brain organoid technology has evolved from rudimentary three-dimensional aggregates into sophisticated models capable of recapitulating human-specific neurodevelopmental processes, including cortical layering, gliogenesis, and functional network formation. Bioengineering innovations, such as vascularized microfluidic systems and stimulation protocols, now enable functional maturation within 60 days (shorten 50% comparing to the conventional culture), achieving electrophysiological synchrony indices > 0.8 [44,57,88]. The advent of innovative culture protocols—characterized by dynamic media supplementation, bioengineered vascularization, and microfluidic perfusion systems—has significantly augmented the longevity beyond 200 days, enabling functional maturation of these organoids. Such advancements empower these constructs to emulate later stages of development, encompassing synaptogenesis, gliogenesis, and network plasticity, thereby bridging critical gaps in the exploration of neurodevelopmental disorders, such as autism [89,90,91] and adult-onset maladies, including Alzheimer’s disease [15,92,93]. Moreover, the incorporation of patient-derived induced pluripotent stem cells (iPSCs) has rendered organoids indispensable instruments for personalized drug screening, as evidenced by their pivotal role in the identification of tauopathy inhibitors and neuroprotective compounds [52,94,95]. While this review concentrates on maturation strategies for spontaneously self-organizing brain organoids derived predominantly from hPSCs, it is important to acknowledge the parallel development of sophisticated mixed cell systems. These systems often involve the deliberate co-culture or assembly of primary neural cells or individually differentiated hPSC-derived neural cell types (e.g., neurons, glia, endothelial cells) to model specific circuits or tissue interactions. Although these engineered multicellular assemblies represent a powerful complementary approach with distinct strengths and limitations, a detailed discussion of their maturation falls outside the scope of this review, which is focused on enhancing the intrinsic developmental trajectory of integrated hPSC-derived organoids.

However, persistent challenges demand integrated solutions: batch-to-batch variability, limited scalability, and the absence of standardized maturity metrics impede cross-study validation and clinical translation. Although beyond the scope of this review, the incomplete maturation and functional validation of non-neuronal components—particularly astrocytes—constitutes a major constraint. Their failure to robustly form essential structures (e.g., glia limitans and functional blood–brain barrier) and the difficulty in defining in vitro reactive states significantly restricts modeling of neurovascular unit biology, neuroimmune interactions, and astrocyte-mediated processes such as edema regulation and amyloid dynamics [26]. A synergistic approach that melds chronological optimization—such as prolonged culture with metabolic support—and active bioengineering techniques, including electrical stimulation, may yield organoids of heightened translational relevance; however, rigorous validation across diverse laboratories remains paramount. To address these gaps, the field must establish unified quantitative frameworks integrating structural, functional, and metabolic parameters. Herein lies the promise of multimodal bioengineering convergence.

Beyond conventional approaches, artificial intelligence (AI) and novel algorithms (e.g., deep learning) are emerging as catalytic forces [96,97,98]. Machine learning (ML) algorithms expand the possibilities to dissect multi-omics landscapes (transcriptomics/proteomics/metabolomics) to pinpoint optimal growth factor cocktails and culture conditions, reducing batch variability. Critically, these computational tools must integrate with real-time, non-invasive monitoring technologies. Hyperspectral imaging, for example, maps metabolic gradients (e.g., lactate/oxygen ratios) to resolve spatial heterogeneity [99], while feedback-driven systems enable automated adjustment of bioengineered inputs (e.g., electrical stimulation intensity) based on neural activity signatures [88]. In the future, the development of non-invasive monitoring techniques may enable real-time and longitudinal studies, empowering researchers to track dynamic cellular and biochemical changes without compromising the viability of the organoids. Such closed-loop intelligence transforms static cultures into dynamically optimized ecosystems.

Moving forward, a synergistic roadmap combining benchmarking frameworks and intelligent engineering is essential (see Figure 3). Establish quantifiable maturity standards, for example, TBR1^+^/SATB2^+^ neuron ratios >80% for cortical layering [31]; MEA-based Synchrony Index > 0.7 for network maturity [20]; ATP/lactate ratio > 2.0 indicates metabolic homeostasis [50]. Deploy AI-navigated bioengineering that adapts stimuli (electrical/magnetic/chemical) to real-time organoid responses [44,78,98]. Advance organoid–humanized interfaces via organ-on-chip neurovascular integration [64] and CRISPR-engineered disease reporters [35]. Develop refined co-culture/assembloid systems integrating iPSC-derived astrocytes, microglia, endothelial cells, and pericytes under optimized biophysical and biochemical cues to promote glia limitans formation and BBB maturation [36].

## 5. Conclusions

This review has navigated the evolving landscape of brain organoid maturation, highlighting two synergistic pillars: the establishment of standardized multimodal benchmarking frameworks and the implementation of innovative bioengineering strategies. The field has made significant strides in accelerating functional maturation through vascularization, microenvironment modulation, and stimulation-based paradigms, effectively halving the time required to achieve electrophysiological and synaptic maturity. Despite these advances, key challenges—including batch variability, astrocyte immaturity, and scalability limitations—persist. The integration of artificial intelligence, real-time monitoring, and refined co-culture systems represents a promising frontier for overcoming these hurdles. Ultimately, the convergence of biologically grounded benchmarks with intelligently engineered maturation accelerators will propel brain organoids from exploratory tools to clinically predictive platforms. To realize their full translational potential, future work must prioritize rigorous validation against human neuropathological data, ensuring these next-generation models faithfully recapitulate the complexity of the human brain across developmental and disease states.

## Figures and Tables

**Figure 1 biomolecules-15-01118-f001:**
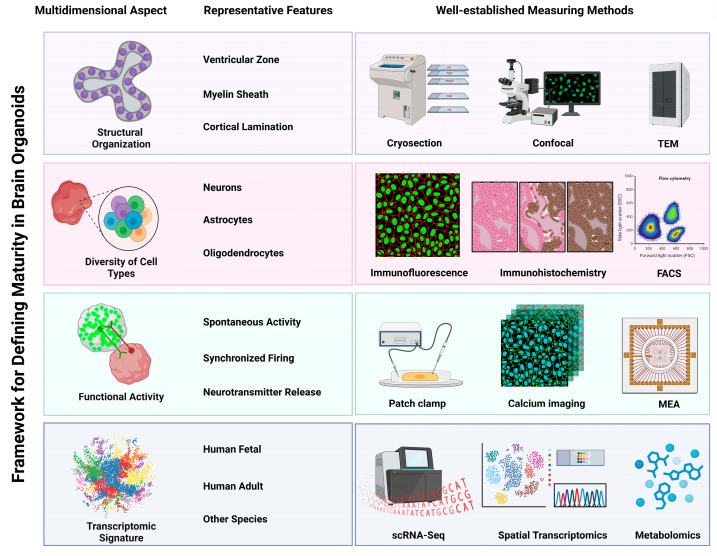
Multidimensional framework for defining maturity in brain organoids. This schematic illustrates key dimensions and methodologies employed for defining maturity in brain organoids. Structural maturation—including ventricular zone formation, myelination, and cortical lamination—is analyzed via cryosectioning, confocal microscopy, and transmission electron microscopy (TEM). Cellular identity is assessed by detecting neurons, astrocytes, and oligodendrocytes using immunofluorescence (IF), immunohistochemistry (IHC), or fluorescence-activating cell sorter (FACS). Functional activity is determined by spontaneous firing, network synchronization, and glutamate release, measured using patch-clamp electrophysiology, calcium imaging, and multielectrode arrays (MEA). Transcriptomic signature is assessed by comparing gene expression profiles to fetal and adult brain references using single-cell RNA sequencing (scRNA-seq), spatial transcriptomics, and metabolomics. Created in BioRender. http://www.biorender.com/.

**Figure 2 biomolecules-15-01118-f002:**
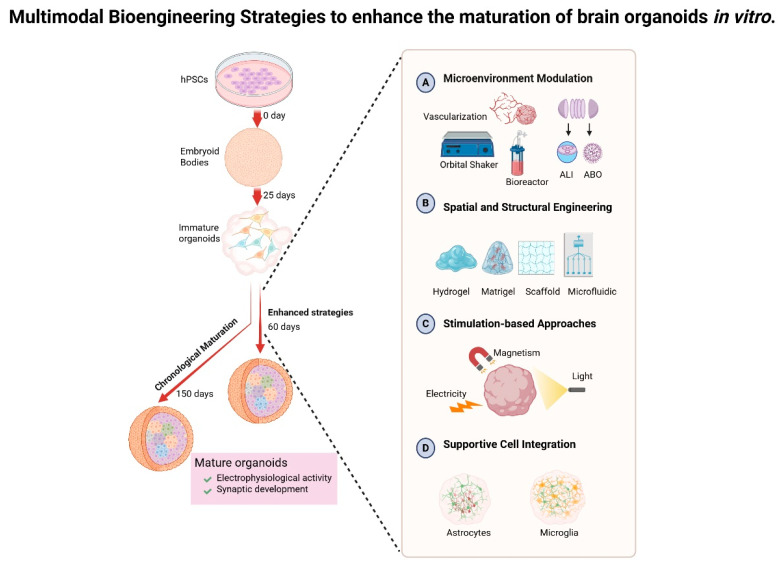
Accelerating brain organoid maturation using bioengineering strategies. hPSCs-derived brain organoids typically require long-term culture (~150 days) to reach partial maturity. Bioengineering approaches—including microenvironment modulation, structural engineering, stimulation, and co-culture with supportive cells—can accelerate maturation (~60 days), enhancing electrophysiological activity and synaptic development. (**A**–**D**) illustrate multimodal bioengineering strategies to enhance brain organoid maturation in vitro. Created in BioRender. http://www.biorender.com/.

**Figure 3 biomolecules-15-01118-f003:**
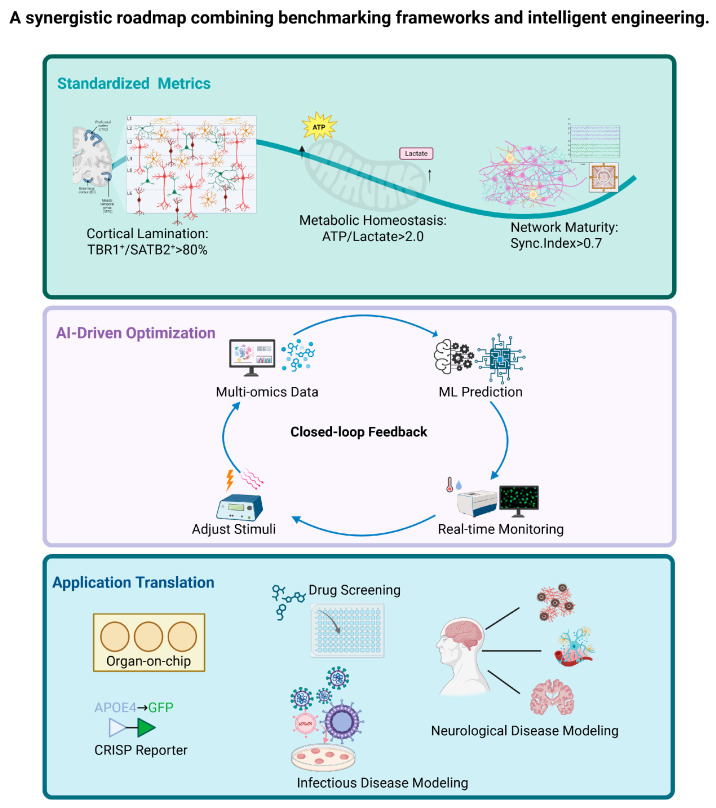
A synergistic roadmap combining benchmarking frameworks and intelligent engineering. In the future, the synergistic convergence of bioengineered innovations—integrating AI-driven optimization, real-time metabolic mapping, and automated culture platforms—will empower the next generation of organoids to transcend current limitations in scalability, reproducibility, and functional maturity. Created in BioRender. http://www.biorender.com/.

## Data Availability

No new data were created or analyzed in this study.

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
