# Peer review of "Navigating Brain Organoid Maturation: From Benchmarking Frameworks to Multimodal Bioengineering Strategies"

_biomolecules, 2025, doi:10.3390/biom15081118_

Round 1
Reviewer 1 Report
Comments and Suggestions for Authors
In this review article, Huang et al. discuss the issues of human brain organoid (hBO) maturation with a view on the evaluation and long-term maturation. The authors discuss the main problems of hBO maturation, such as the need of prolonged culture and the degeneration of the organoid’s core due to lack of nutrients or oxygenation. In the first part, the benchmarks and methods of assessment of maturation were reviewed, followed by the discussion of engineering strategies to facilitate and accelerate the maturation of the hBOs, such as vascularization, microenvironment modulation, stimulation strategies, etc.
The review is clearly written and easy to read. Only one typo was detected by this reviewer (line 83; “Multimensional” should be corrected to “Multidimensional”), but another spell-check is recommended to make sure that there are no errors. Regarding the reader base, novices as well as researchers that have experience in this field would find this review very helpful.
No major problems were found with the review, and the recommendation is to be accepted after minor grammar check.
Author Response
Comments 1: A typo was detected on line 83; “Multimensional” should be corrected to “Multidimensional”.
Response 1: Thank you for pointing out this typo. We have corrected “Multimensional” to “Multidimensional” on page3, line 92 in the latest version of the manuscript.
Comments 2: A recommendation for another spell-check to ensure there are no errors.
Response 2: We appreciate your suggestion for an additional spell-check. We have conducted a thorough review of the manuscript and have corrected any remaining typographical errors to ensure clarity and accuracy.
Reviewer 2 Report
Comments and Suggestions for Authors
This review highlights progress of brain organoids as an extension and alternative to enhance "in vitro modeling of human neurodevelopment and disease". The authors address advantages of human 3D brain tissue organoids over the limitations of rodent models for questions of CNS development. They address the desired properties of laminar relationships between layers of brain tissue and cell type development and diversity seen with in vivo brain regional architectures. They also balance the discussion by addressing the constraints of 3D brain tissue organoid for the resource intensive and challenging length of time in tissue culture required for many slowly developing neurological features, diseases and drug therapeutic effects. The authors address these gaps by pointing to systems that engineer bridging solutions with "more sophisticated culture systems, bioreactors, and bioengineering approaches that can better mimic the environment in vivo" to compress the time frames in which 3D constructs may adequately ask and answer focused questions with in vitro human tissue systems. The authors also address the desire for improving the field with "standardized maturity metrics" that this field currently lacks. The review is timely, well documented from the literature, and presents a fair and balanced state of the art for where 3D brain organoid science has been, where it is now, and how it can move forward in vitro as a flexible system of tools for neurologic sciences that may well offer solutions for other organ systems and disease modeling in medical sciences.
Major concerns:
none
Minor concerns:
- Line 161 should "practically" be "practicality"?
- Consider if italics should be used (traditionally) for in vitro, in vivo Latin terms.
- The review essentially focuses on iPSC derived spontaneous brain organoid development from the literature and does not specifically include mixed cell systems from primary and/or individual iPSC derived brain cells. Frankly, this seems fine for this review as the latter is equally complex and has its own set of strengths and limitations. At most, a mention by a sentence or 2 that those data are not covered here in detail will inform the reader that there are also many manuscripts that take a deliberate multicellular design approach, vs these mostly iPSC derived larger brain organoids. Both add value to the literature, and this review is fine for the depth and especially the breadth with which it discusses the breadth and depth of the spontaneous organoids and engineering principles that may enhance future work.
Author Response
Comments 1: Line 161 should "practically" be "practicality"?
Response 1: Thank you for pointing this out. We have corrected "practically" to "practicality" in page5, line 183 in the latest version of the manuscript.
Comments 2: Consider if italics should be used (traditionally) for in vitro, in vivo Latin terms.
Response 2: We appreciate your suggestion. We have italicized the terms "in vitro" in line 18, 54, 90, 207, 241, 261, 266-267, 273, 291, 435, 459, 495; and "in vivo" in line 82, 271, 272, 326, 335, 425 in the latest version of the manuscript throughout the manuscript to adhere to traditional formatting conventions.
Comments 3: The review essentially focuses on iPSC derived spontaneous brain organoid development from the literature and does not specifically include mixed cell systems from primary and/or individual iPSC derived brain cells. Frankly, this seems fine for this review as the latter is equally complex and has its own set of strengths and limitations. At most, a mention by a sentence or 2 that those data are not covered here in detail will inform the reader that there are also many manuscripts that take a deliberate multicellular design approach, vs these mostly iPSC derived larger brain organoids. Both add value to the literature, and this review is fine for the depth and especially the breadth with which it discusses the breadth and depth of the spontaneous organoids and engineering principles that may enhance future work.
Response 3: We acknowledge this point and have added a brief mention in the ‘Conclusion and Perspectives’ section to clarify that while mixed cell systems and primary cell-derived organoids are valuable and complex, this review primarily focuses on human pluripotent stem cell (hPSC)-derived brain organoids. This addition aims to inform readers of the existence and significance of alternative multicellular design approaches in the literature.
Page11-12, line 453-462: “While this review concentrates on maturation strategies for spontaneously self-organizing brain organoids derived predominantly from hPSCs, it is important to acknowledge the parallel development of sophisticated mixed cell systems. These systems often involve the deliberate co-culture or assembly of primary neural cells or individually differentiated hPSC-derived neural cell types (e.g., neurons, glia, endothelial cells) to model specific circuits or tissue interactions. Although these engineered multicellular assemblies represent a powerful complementary approach with distinct strengths and limitations, a detailed discussion of their maturation falls outside the scope of this review, which is focused on enhancing the intrinsic developmental trajectory of integrated hPSC-derived organoids.”
Reviewer 3 Report
Comments and Suggestions for Authors
I appreciate the enthusiasm of the authors for the role of CNS human organoids in progress in Neuroscience.
My critique focuses on limitations of this in vitro methodology in relation to modelling mechanisms and their manipulation: pharmacologic and otherwise.
This review paper would benefit from identification and discussion of limitations with, perhaps, providing directions as how to address them and thus improve the functionality of CNS organoids.
The primary focus should be on astrocytes as cells that control and regulate the CNS in health and in disease. Here are some select but not all questions to consider:
- do astrocytes in human CNS organoids form glia limitans?
- does blood-brain barrier form in human CNS organoids?
- what is the status of activation of astrocytes in organoids cultured in standard conditions? How do you characterize "activated" astrocytes in organoids?
- can you study, how?, reactive functions of astrocytes, for example: anti-inflammatory?, anti-edema? astrocytic erythropoiesis? metabolism of amyloid?
- etc.
Ultimately, improved CNS organoids should come under scrutiny of neuropathologists prior to their acceptance as an accepted method in developmental process of regulatory treatment approval.
-
Author Response
We sincerely appreciate your insightful critique regarding the limitations of CNS organoid models, particularly the need to critically address astrocyte functionality and non-neuronal components. In direct response to your recommendations, we have implemented the following substantive revisions throughout the manuscript, with all modifications highlighted in blue text for your convenience. Please see the attachment.
